# An IT Platform Supporting Rectal Cancer Tumor Board Activities: Implementation Process and Impact Analysis

**DOI:** 10.3390/ijerph192315808

**Published:** 2022-11-28

**Authors:** Maria Lucia Specchia, Andrea Di Pilla, Maria Antonietta Gambacorta, Alessandro Filippella, Flavia Beccia, Sara Farina, Elisa Meldolesi, Chiara Lanza, Rocco Domenico Alfonso Bellantone, Vincenzo Valentini, Giampaolo Tortora

**Affiliations:** 1Fondazione Policlinico Universitario A. Gemelli IRCCS, 00168 Rome, Italy; 2Department of Life Sciences and Public Health, Università Cattolica del Sacro Cuore, 00168 Rome, Italy

**Keywords:** rectal cancer, multidisciplinary tumor boards, IT platform, implementation, impact analysis, digital health

## Abstract

Colorectal cancer (RC) is the third most common cancer, with an increasing incidence in recent years. Digital health solutions supporting multidisciplinary tumor boards (MTBs) could improve positive outcomes for RC patients. This paper describes the implementation process of a digital solution within the RC-MTB and its impact analysis in the context of the Fondazione Policlinico ‘A. Gemelli’ in Italy. Adopting a two-phase methodological approach, the first phase qualitatively describes each phase of the implementation of the IT platform, while the second phase quantitatively describes the analysis of the impact of the IT platform. Descriptive and inferential analyses were performed for all variables, with a *p*-value < 0.05 being considered statistically significant. The implementation of the platform allowed more healthcare professionals to attend meetings and resulted in a decrease in patients sent to the RC-MTB for re-staging and further diagnostic investigations and an increase in patients sent to the RC-MTB for treatment strategies. The results could be attributed to the facilitated access to the platform remotely for specialists, partly compensating for the restrictions imposed by the COVID-19 pandemic, as well as to the integration of the platform into the hospital’s IT system. Furthermore, the early involvement of healthcare professionals in the process of customizing the platform to the specific needs of the RC-MTB may have facilitated its use and contributed to the encouraging quantitative results.

## 1. Introduction

Colorectal cancer is the third most common cancer, the fourth leading cause of death and the second leading cause of cancer death worldwide [1,2]. Rectal cancer (RC) accounts for about one-third of colorectal cancer cases, with an increasing incidence in individuals under 50 years both in the USA and Europe [1]. 

Nevertheless, over the last few years, clinical outcomes in RC care have significantly improved mainly due to the advancements in different specialties such as surgery, oncology, radiology and pathology [3]. The RC clinical management is evolving, and the appropriateness of surgical treatment options for selected patients and the use of multimodality treatment pathways could significantly influence both recurrence and survival. Therefore, modern treatment strategies are based on the close working of related disciplines within multidisciplinary teams [1], and multidisciplinarity, which is recognized as an effective approach to improve healthcare quality and patient outcomes in cancer care [4,5], is increasingly a fundamental requirement for the appropriate management of RC patients [3]. Indeed, the positive impact of the multidisciplinary approach and the multidisciplinary tumor boards (MTBs) on the outcome of RC patients is described in the scientific literature [3].

MTBs are multidisciplinary meetings focused on oncological patient care [4,6]. They are also referred to as Multidisciplinary Cancer Conferences (MCC), which are forums for healthcare providers aimed at discussing the diagnostic and treatment of cancer patients [7,8]. MTBs include different specialties of healthcare professionals (surgeons, oncologists, radiologists, pathologists, nurses, etc.) depending on factors such as type of cancer presented and discussed and hospital size [4]. In particular, different levels of involvement of team members have been described: core and allied members, which are all medical specialists, and support members including both medical and non-medical professionals [9].

The MTBs working format consists of scheduled daily or weekly meetings, in person, or through digital health solutions such as video or teleconferencing [4]. Digital health is a broad scope embracing tools and methodologies such as health information technology, telehealth and telemedicine, mobile health, wearable devices, and personalized medicine [10,11]. The implementation of digital health solutions supporting MTBs is part of a considerable transformation that the healthcare sector is undergoing all over the world due to the significant adoption of digital technologies to empower patients and professionals in reaching health goals [12,13]. In 2019, “Recommendations on digital interventions for health system strengthening” have been released by the World Health Organization (WHO), providing a guidance for the greater use and integration of technology for healthcare progress [14]. Moreover, the COVID-19 pandemic is accelerating the healthcare digital transition due to the rapid adoption of advanced digital solutions and technology tools by healthcare organizations in response to the global crisis to mitigate its impact on both individuals and health systems [15,16]. In clinical practice, in the last few years, several digital platforms have been studied and implemented for the management of treatment and well-being of cancer patients [17,18], or in the management of other non-communicable diseases [19,20,21,22], striving for safety and quality of care [23]. However, the adoption of digital tools in patient management within multidisciplinary teams is something that is still little considered [24]. Building on the potential of this application, arising from the synergistic combination of the benefits of both digital tools and the interplay between specialists from different disciplines, as explained above, this paper reports on the experience of the RC-MTB of the Comprehensive Cancer Centre (CCC) of the Fondazione Policlinico “A. Gemelli” (FPG) Scientific Research and Care Institute (IRCCS), in Rome, Italy, with the double aim to: (a) describe the implementation process of a digital solution within the RC-MTB; and (b) evaluate its impact on MTB activity volumes and organizational and patient-related aspects during the COVID-19 pandemic.

## 2. Materials and Methods

This study is compliant with the Local Ethical Committee Standards of the FPG IRCCS. It was carried out in accordance with the Helsinki Declaration and EU Regulation 2016/679 (GDPR). 

We adopted a two-phase methodological approach, consisting of a qualitative description of the IT platform implementation and of the quantitative impact analysis, hereby described.

### 2.1. Phase 1: IT Platform Implementation

Starting from January 2020, the CCC of the FPG IRCCS launched a program for the adoption of an IT platform to support the activities of MTBs with the aim of standardizing, organizing, and managing the MTBs’ activities particularly regarding multidisciplinary discussion of clinical cases, shared decisions on the clinical path and its implementation.

In the original process, cases identified as deserving of multidisciplinary investigation were proposed periodically to the boards, which met in a meeting room. The discussion of cases was supported mainly by PowerPoint presentations, which were prepared before the meeting for each clinical case to be discussed, “manually” reporting the patient’s clinical data and info and the results of laboratory and instrumental examinations. Each meeting was corresponded by the preparation of a paper record with the collection of attendance, meeting summary and clinical decisions.

The IT platform implementation process, within the RC-MTB, consisted of the following steps:Creation of the multidisciplinary working group (WG) for the platform customization according to the RC-MTB specific needs;Presentation of the IT platform to the WG;Definition of the data to be recorded and stored in the platform for each clinical case discussed;Customization of the platform according to the RC-MTB needs;Platform integration into the hospital IT system to allow the automatic import of patients’ data;Definition by the WG of the activity flow for the platform integration into the RC-MTB’s activities.

The results of each step were used to revise the prototype. Subsequently, the platform was validated through a pilot test and operatively launched. 

### 2.2. Phase 2: IT Platform Impact Analysis 

The platform impact analysis was carried out through a pre/post comparison of the RC-MTB in terms of activity volumes and organizational aspects (number of MTB meetings, number of cases discussed during each session, number of healthcare professionals involved in each meeting) and patient-related aspects (clinical questions for the MTB discussion, cancer stage and shared clinical decision). For the period prior to the implementation of the app, 2019 data were collected and analyzed, while for the period after, 2020 data were used. Respectively, 2019 data were extracted from paper records, while 2020 data were extracted from the IT platform. All the data were collected in an Excel database.

The possible clinical questions for RC-MTB discussion and shared clinical decisions were categorized as shown in Table 1.

For all the variables considered, descriptive and inferential analyses were carried out. For descriptive analysis, absolute and relative frequencies and means were calculated for each year. For the inferential analysis, Student’s *T*-test and Chi-Square test were performed. Results with a *p*-value < 0.05 were considered statistically significant.

## 3. Results

### 3.1. Phase 1: IT Platform Implementation

The platform integration with the institutional information flows allows the automatic import of the clinical reports of laboratory and histopathological diagnostic tests and diagnostic imaging.

The activity flow for the IT platform implementation to support the RC-MTB involves the following sequential steps, which are described in detail in Table 2: Request for multidisciplinary assessment of a patient by RC-MTB;Collection of clinical information/data and creation of an online folder for each patient to be discussed at RC-MTB;Schedule of RC-MTB on the platform;RC-MTB conference on the platform;Report of the conference and “TO DO list”;“TO DO list” implementation;Report of the completed tasks on the platform.

### 3.2. Phase 2: IT Platform Impact Analysis

Concerning RC-MTB activity volumes and organizational aspects, in 2019-2020, a total number of 836 cases and 560 patients were discussed (patients can be discussed at MTB more than once) in 88 RC-MTB meetings. The pre/post difference of cases discussed on average in each meeting was not statistically significant (*p* = 0.07). However, in 2020, more healthcare professionals participated on average in each meeting compared to 2019, with a statistically significant difference between the two years (*p* < 0.001) (Table 3).

Concerning patient-related aspects, results are reported in Table 4.

In 2019, patients were mostly referred to RC-MTB for “primary staging and re-staging after radiotherapy-chemotherapy” (36.7% and 33.1% of cases, respectively). In 2020, the main clinical questions were “treatment strategy” (38.3%) and “primary staging” (34.8%). Comparing each clinical question between 2019 and 2020, the decrease in patients referred to RC-MTB for “re-staging after radiotherapy-chemotherapy” (from 33.1% to 22.9%) and the increase in patients referred to RC-MTB for “treatment strategy” (from 24.1% to 38.3%) were statistically significant (*p* = 0.001 and *p* < 0.001, respectively).

The majority of patients discussed at RC-MTB had a stage III cancer (65.1% and 70.3% in 2019 and 2020, respectively). No statistically significant differences were found as regards cancer stage between 2020 and 2019.

The most frequent shared clinical decision in 2019 was “needed further diagnostic investigation results” (32% of cases), which showed a significant reduction (to 24.9% of cases) in 2020 (*p* = 0.023). The most frequent shared clinical decision in 2020 was “referring patients for RT-CT” (31.6% of cases) with a statistically significant increase compared to 20.6% of cases in 2019 (*p* < 0.001). 

## 4. Discussion

The results we obtained from our study on the IT platform adoption showed a statistically significant improvement in the operation of the RC-MTB in 2020 compared with 2019, when it was not yet used. Our results showed a positive impact of the platform for both organizational and patient-related aspects. 

An increase in the number of registered participants at meetings was observed, from an average of 5.1 in 2019 to 12.2 in 2020. This result could be attributed to the facilitated access to the platform for specialists having more difficulty attending meetings in person or, alternatively, to the greater reliability of online reporting of attendance compared to paper documentation. In addition, it should be noted that the opportunity offered by the platform to participate in meetings remotely certainly played a key role in compensating for the restrictions imposed by the COVID-19 pandemic, ensuring continuity to the RC-MTB activities. Indeed, as reported by Al-Shamsi HO et al., 2020 and Paterson C et al., 2020, the digital approach also helps to smooth organizational technicalities and limitations to in-person meetings, providing a simple, rapid and accessible solution for all the TB participants [25,26].

The platform also allowed an increased number of patients to be discussed in 2020, implying that its use was not an obstacle to the admission of more patients to RC-MTB, considering the need, for each patient to be discussed, to collect clinical information and create an online folder. This was plausibly due to the platform integration into the hospital IT system and the automatic import of patients’ data, which make it possible to optimize the preparative work for the discussion and also evaluate more clinical cases in less time, which can also leave more time to discuss highly complex patients. This is in line with what is described in the literature. Indeed, the data-banking system and virtual meetings have been reported by Blasi et al. 2021 to be time-saving [24]. In addition, the Care Manager, as a dedicated professional figure, assumes a key role in the collection of the documentation to be examined and analyzed during the multidisciplinary meeting, the data entry and reporting of patient’s clinical history. Indeed, the centralization of these activities guarantees the appropriateness and completeness of clinical information and data without fragmentation of responsibilities [27,28].

Moreover, the platform proved to have the potential to improve the programming of RC-MTB discussions by significantly impacting the reduction in the proportion of ‘premature’ cases, i.e., cases for which the shared clinical decision was “needed further diagnostic investigation results”, and therefore, further diagnostic tests had to be requested before discussion. As a consequence, predominantly cases for which diagnostic data were already available (e.g., derived from endoscopic, biopsy, CT or MRI examinations or nuclear medicine) were brought to the multidisciplinary discussion. This result was certainly due to the higher control allowed by the software on the completeness of the diagnostic examinations needed for the clinical decision [29].

Concerning patients’ characteristics, our results showed that the majority of cases arrived at TB meeting in stage III (65.1% in 2019 and 70.3% in 2020), without statistically significant changes in the period 2019–2020. This is not surprising, since we do not expect that MTB impacts on cancer stage at the time of diagnosis (which is mostly influenced by primary prevention and screening), but rather on clinical decision making and, therefore, on clinical outcomes. Indeed, scientific evidence shows that promoting a digital aid for MTB strengthens the health professionals’ network and the multidisciplinary approach to the decision-making process to provide the best cancer care by improving patient management plans thanks to the combined knowledge of different specialists [24]. As regards clinical decision making, our study showed a significant increase in the percentage of patients undergoing radio-chemotherapy. This is partly a result of the platform implementation, which allowed the immediate availability of clinical information, improving data accessibility and usability, and promoted multidisciplinary discussion between specialists. The synergistic effect of these aspects reasonably may have favored the management of patients who, despite being at an advanced stage of the disease, could benefit from cycles of radio-chemotherapy. In addition, it should also be considered that the end of 2019 and the beginning of 2020 saw a significant impulse for use of MRI-LINAC, the magnetic resonance imaging with linear accelerator, which allowed many more patients to be treated, especially patients with advanced cancer [30].

Concerning reasons behind the success of the IT platform implementation process, the platform usability, in terms of data retrieval and interface easiness, is likely to have facilitated its adoption by the various specialists involved in MTB [31]. Moreover, the early involvement of the ‘end-users’, i.e., the medical professionals, in the platform customization process according to the RC-MTB specific needs (Phase 1) may have benefited its usability and contributed to the encouraging quantitative results (Phase 2) [32]. Lastly, the close collaboration between the ‘end-users’ and the IT support staff addressed the need for engagement and the involvement of complementary competence profiles in digital health [33].

The positive impacts reported for the IT platform applied to RC-MTB activities prove the usefulness and benefits of integrating digital tools in multidisciplinary cancer care. According to the available literature, the multidisciplinary strategy, through the use of digital technology, results in an improved workflow efficiency [34] with also a potential direct economic impact on hospitals [35] and a better outcome for cancer patients both in the short and long term alongside an improved experience of diagnostic and therapeutic pathways in oncology [5,24].

The program for the adoption of a digital tool to support the MTBs within the FPG IRCCS stands as a virtuous process aimed at standardizing activities and promoting professional integration and multidisciplinary approach in cancer care, the effectiveness of which is widely recognized [1,4].

For future perspectives, on the clinical management side, firstly, it would be of interest to expand the assessment of virtual MTBs impact on a larger case series and to other cancer diseases, and to analyze it not only in terms of activity volumes but also in terms of processes (e.g., timing of clinical paths and adherence to guidelines), clinical outcomes for patients (such as survival and quality of life), organizational outcomes for healthcare professionals and healthcare providers. Secondly, virtual MTBs offer a tool to integrate professionals also from different geographical areas, allowing for high-quality treatment recommendations by expanding the local MTB team into a network of national/international experts, to support multidisciplinary cancer care also in settings where funding and resources may be limited [36]. A further perspective in the near future could be the systematic use of the IT platform, as well as to support clinical management, also for research and teaching activities. The platform implementation could benefit from the integration of Artificial Intelligence tools [37], such as the adoption Bayesian networks for digital patient models in helping the MTB decision-making process [38] or the multifactorial Decision Support System in elaborating predictive models by the integration of all available data (clinical, imaging, biologic, genetic, cost, etc.) [39]. AI-based medical article retrieval and integrated analysis of Electronic Health Records data using AI are also aspects that could be evaluated in the future developments of the platform [37].

This work must be considered in light of certain limitations. The data analyzed cover a limited time span, without assessing the sustainability of the platform and its effectiveness in cancer care beyond the chosen indicators. The study did not analyze other aspects related to the implementation of the platform, such as the quantitative impact in terms of resources used and outcomes for healthcare providers and healthcare professionals. Moreover, any additional analysis was not performed to identify potential confounding factors, related to hospital organization, technologies or processes, that could have influenced the adoption of the tool and the results.

## 5. Conclusions

The COVID-19 pandemic has provided a spur on the implementation of digital tools, due to the restrictions imposed, and has facilitated the adoption of these tools, examples of which have multiplied in recent years, thanks to technological and scientific progress.

The promising results of this study, reporting the implementation process and the impact analysis of the IT platform adopted, are in line with what is found in the literature. This aspect, considered in combination with the scalability of the tool, paves the way for the digital transformation of MTBs with the ultimate goal of increasing the quality and safety of cancer care. Involving healthcare professionals in the early steps of the implementation process shows benefits for healthcare professionals and patients. In conclusion, reporting the implementation process and the impact analysis of the IT platform adopted could provide useful tools for healthcare management and services and incentivize both MTBs and digital health solutions. 

## Figures and Tables

**Table 1 ijerph-19-15808-t001:** Clinical questions for RC-MTB discussion and shared clinical decisions.

Code	Definition
*Clinical Questions for RC-MTB Discussion*	
1. Primary staging	patients for whom a primary staging has to be defined or discussed (cTNM)
2. Re-staging after radiotherapy-chemotherapy	patients for whom a new staging has to be defined after neoadjuvant treatments (radiotherapy and/or chemotherapy) (ycTNM)
3. Post-surgical re-evaluation	patients for whom a pathological staging has to be defined upon the histopathological examinations of operation findings (pTNM or ypTNM)
4. Treatment strategy	patients for whom a treatment course (or new treatment course) is to be shared in view of the availability of new information, e.g., patients who show possible signs of progression on CT or MRI, or in general any patient for whom diagnostic doubt arises or regarding appropriate treatment protocols
5. Post-recurrence evaluation	patients who are re-evaluated after imaging examinations have shown definite signs of recurrence (usually after a course of therapy now completed)
*Shared clinical decision*	
1. Follow-up	patients who are prescribed/recommended to return after a pre-defined time interval for scheduled new examinations according to clinical guidelines/protocols
2. Needed further diagnostic investigation results	patients who cannot be properly evaluated on the basis of the available exams results and who need further examination (biopsy or imaging) to draw a definitive conclusion on diagnosis; therefore, they are postponed to later MTB sessions
3. Surgery	patients are referred for surgical treatment
4. CT	patients are referred for chemotherapy
5. RT	patients are referred for radiation therapy
6. RT-CT	patients are referred for radio-chemotherapy

**Table 2 ijerph-19-15808-t002:** Activity flow for the IT platform implementation.

Step	Description
Request for multidisciplinary assessment of a patient by RC-MTB	The Referring Doctor, that is the doctor in charge of the patient, fills in the application form for the request of the multidisciplinary assessment of the patient and sends it by e-mail to the e-mail box of the RC clinical pathway.
2. Collection of clinical information/data and creation of an online folder for each patient to be discussed at RC-MTB	After receiving and reading the request, the Care Manager (CM), who is the referring nurse, responsible for the coordination of the whole clinical pathway’s activities, from patient’s taking charge to follow-up, collects the documentation needed for the discussion, according to the RC clinical pathway. At this stage, the CM, by accessing the Teaching Hospital’s registry through the IT platform, can find and automatically import all the reports of the clinical examinations and diagnostic imaging the patient underwent within the hospital, to be examined and analyzed during the multidisciplinary meeting. The CM can also upload directly to the platform any exams stored on DVD/CD carried out in other hospitals and taken by the patient. Finally, the CM fills in the appropriate MTB’s data sheets configured in the platform by both the data entry and a short free text reporting the patient’s clinical history. Therefore, for each patient to be discussed at MTB, an online folder containing all the clinical documentation of interest is created on the platform. The RC-MTB Coordinator, who is the doctor in charge of the MTB, checks all the documentation to ensure completeness and appropriateness of the collected information and clinical questions.
3. Schedule of RC-MTB on the platform	The CM schedules on the platform the multidisciplinary discussion of the patient and sends, by an email generated by the platform, all the MTB members the invitation to the multidisciplinary meeting.
4. RC-MTB conference on the platform	The RC-MTB takes place with the participation, in person or by teleconferencing, of the professionals involved and the IT platform supporting all the discussion steps. The simultaneous online consultation of the patient’s medical history, data and clinical examination reports and the imaging display allows the assessment and discussion of cases by the MTB members to reach the definition of shared care plans.
5. Report of the conference and “TO DO list”	At the end of the meeting, the CM, through the platform interface, generates the RC-MTB minute and the “TO DO list”, that is the list of actions to be carried out to follow up the care plans shared and defined by the RC-MTB. The RC-MTB Coordinator checks all the documentation to ensure completeness and appropriateness of the collected information and clinical questions.
6. “TO DO list” implementation	The CM takes charge of the implementation of the “TO DO list” (booking of further clinical examinations, listing a patient for hospitalization/surgery, scheduling medical therapy, and providing the patient/caregiver with all appropriate information/communications).
7. Report of the completed tasks on the platform	The CM gradually records on the platform the activities completed.

**Table 3 ijerph-19-15808-t003:** RC-MTB activity volumes and organizational aspects.

	2019	2020	*p*-Value
	*N* (%)	*N* (%)	
Cases discussed	390 (47%)	446 (53%)	
Patients discussed	246 (44%)	314 (56%)	
MTB meetings	38 (43%)	50 (57%)	
	** *Mean* **	** *Mean* **	
Cases discussed per meeting	10.3	8.9	0.07
Healthcare professionals involved in each meeting	5.1	12.2	<0.001 *

* *p*-value < 0.05.

**Table 4 ijerph-19-15808-t004:** RC-MTB patient-related aspects.

		2019	2020	*p*-Value
		*N* (%)	*N* (%)	
*Clinical questions*	Primary staging	143 (36.7)	155 (34.8)	0.564
	Re-staging after radiotherapy-chemotherapy	129 (33.1)	102 (22.9)	**0.001**
	Post-surgical re-evaluation	8 (2.1)	4 (0.9)	0.162
	Treatment strategy	94 (24.1)	171 (38.3)	<0.001 *
	Post-recurrence evaluation	16 (4.1)	14 (3.1)	0.455
*Cancer stage ***	I	11 (10.4)	7 (7.7)	0.514
	II	10 (9.4)	14 (15.4)	0.203
	III	69 (65.1)	64 (70.3)	0.434
	IV	16 (15.1)	6 (6.6)	0.050
*Shared clinical decision ***	Follow-up	15 (4.0)	24 (5.4)	0.341
	Needed further diagnostic investigation results	121 (32.0)	111 (24.9)	0.023 *
	Surgery	119 (31.5)	123 (27.6)	0.22
	CT	17 (4.5)	15 (3.4)	0.401
	RT	28 (7.4)	32 (7.2)	0.898
	RT-CT	78 (20.6)	141 (31.6)	<0.001 *

* *p*-value < 0.05; ** Percentages were calculated net of missing data. *p*-value was calculated on the difference between the two years considered.

## Data Availability

The data presented in this study are available on request from the corresponding author.

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
