# Peer review of "An IT Platform Supporting Rectal Cancer Tumor Board Activities: Implementation Process and Impact Analysis"

_ijerph, 2022, doi:10.3390/ijerph192315808_

Round 1
Reviewer 1 Report
Line 34:Introduction Part:
1. The related research is not rich enough. Supplement some references, such as other studies using relevant IT technologies to support treatment.
2. Compared with previous studies, what are the main features of your research?
Line 85:Phase 1: IT platform implementation:
3. Whether the IT platform implementation process is different from the original process should be explained
Line 138: IT platform impact analysis
4. The changes brought by the IT platform to RC-MTB include many aspects. This part currently analyze RC-MTB activity volumes and organizational、patient-related aspects. More comprehensive analysis should be added, such as healthcare providers, healthcare professionals, etc.
Line 174:4. Discussion
5. Ihe limitations of the study are written, and it is suggested that the limitations of the study be written in a separate part.
Reviewer 2 Report
This paper discusses the change made by adapting teleconference system in RC-MTB. I think that this is a good paper since readers can understand the preparation procedure of RC-MTB. But more information of built IT system must be provided for clear understanding and publication.
Specific comments.
1. Bulit-IT platform can include artificial intelligence (AI) system. Role of Care manager will be replaced by AI system in the future. System development discussion of IT platform in the perspective of AI can be include in this manuscript.
2. Previous system without IT platform must be described for clear comparison between 2019 and 2020.
3. You can provide cost and man-power for construction and operation of IT platform.
Round 2
Reviewer 2 Report
This manuscript responded to all my comments. I recommend this paper for publication.